# What is the "normal" fetal heart rate?

Stephanie Pildner von Steinburg[1], Anne-Laure Boulesteix[1,2,5], Christian Lederer[2], Stefani Grunow[3], Sven Schiermeier[4], Wolfgang Hatzmann[4], Karl-Theodor M. Schneider[1] and Martin Daumer[2,3]

[1] Frauenklinik und Poliklinik der Technischen Universität München, Munich, Germany
[2] Sylvia Lawry Centre for Multiple Sclerosis Research e.V., Munich, Germany
[3] Trium Analysis Online GmbH, Munich, Germany
[4] Frauenklinik, Universität Witten, Witten-Herdecke, Germany
[5] Ludwig Maximilians University Munich, Munich, Germany

## ABSTRACT

**Aim.** There is no consensus about the normal fetal heart rate. Current international guidelines recommend for the normal fetal heart rate (FHR) baseline different ranges of 110 to 150 beats per minute (bpm) or 110 to 160 bpm. We started with a precise definition of "normality" and performed a retrospective computerized analysis of electronically recorded FHR tracings.

**Methods.** We analyzed all recorded cardiotocography tracings of singleton pregnancies in three German medical centers from 2000 to 2007 and identified 78,852 tracings of sufficient quality. For each tracing, the baseline FHR was extracted by eliminating accelerations/decelerations and averaging based on the "delayed moving windows" algorithm. After analyzing 40% of the dataset as "training set" from one hospital generating a hypothetical normal baseline range, evaluation of external validity on the other 60% of the data was performed using data from later years in the same hospital and externally using data from the two other hospitals.

**Results.** Based on the training data set, the "best" FHR range was 115 or 120 to 160 bpm. Validation in all three data sets identified 120 to 160 bpm as the correct symmetric "normal range". FHR decreases slightly during gestation.

**Conclusions.** Normal ranges for FHR are 120 to 160 bpm. Many international guidelines define ranges of 110 to 160 bpm which seem to be safe in daily practice. However, further studies should confirm that such asymmetric alarm limits are safe, with a particular focus on the lower bound, and should give insights about how to show and further improve the usefulness of the widely used practice of CTG monitoring.

Corresponding author
Martin Daumer,
daumer@slcmsr.org

## INTRODUCTION

Recording of fetal heart rate (FHR) via cardiotocography (CTG) monitoring is routinely performed as an important part of antepartum and intrapartum care. However, in several randomized trials it became evident that there is only limited efficacy in improving fetal outcome using CTG antenatally (*Pattison & McCowan, 2004*). A detailed meta-analysis of available studies on the use of intrapartum cardiotocogram showed reduction of perinatal

mortality by 50%, but an increase of operative intervention by factor 2.5 (*Vintzileos et al., 1995*). One potential reason is the wide variability in clinical decision making associated with its use. Standardizing management of variant intrapartum FHR tracings was suggested to reduce this variability and to lead to improvement in fetal outcome (*Downs & Zlomke, 2007*). In a recent Cochrane review no difference in outcome could be found when looking at potential improvements through the use of CTG monitoring, but, remarkably, the conclusion was different when computerized interpretation of CTG traces was taken into account: "when computerized interpretation of the CTG trace was used, the findings looked promising" (*Grivell et al., 2012*). Therefore it seems natural to assume that further work on improving definitions and standardization by using computerized methods will further improve the monitoring systems. However, currently, there is not even agreement on the normal range of the baseline of the FHR, although, as Massaniev stated in 1996, "baseline rate provides valuable information on which we plan our further actions" (*Manassiew, 1996*).

The current international guidelines of the Fédération Internationale de Gynécologie et d'Obstétrique (FIGO) (*Rooth, Huch & Huch, 1987*), based on consensus during the 1985 conference, recommend a normal range of the FHR from 110 to 150 beats per minute (bpm). The FIGO guidelines, despite some well-known shortcomings, "remain the sole broad international consensus document in FHR monitoring" (*Diogo & Joao, 2010*). This consensus replaced the former range of 120 to 160 bpm, as there was evidence pointing to worse fetal outcome for baselines higher than 160 bpm (*Saling, 1966*). Up to now, ranges such as 110 to 150 bpm or 110 to 160 bpm (*American Congress of Obstetricians and Gynecologists, 2009*; *Deutsche Gesellschaft für Gynäkologie und Geburtshilfe, 2010*; *Macones et al., 2008*; *Manassiev et al., 1998*; *National Institute for Health and Clinical Excellence (NICE), 2007*; *Perinatal Committee of the Japan Society of Obstetrics and Gynecology, 2009*; *Royal Australian and New Zealand College of Obstetricians and Gynaecologists, 2006*; *Society of Obstetrics and Gynaecologists of Canada, 2007*) are also used, widely based on expert opinion rather than evidence.

This assessment of the situation and the existing "evidence base" is based on the following elements. We have published the plan to do the analysis and have publicly asked for feedback. We have done several literature searches mostly in Pubmed, Google Scholar, the Cochrane Library and have collected publications listed in various versions of published CTG guidelines and standard textbooks. In total we have collected more than 100 papers related to the topic. We have asked opinion leaders in Germany, the UK and the US about awareness of any recent and ancient work that would need to mentioned. In addition, stimulated by the reviewer's comments, we have (March 2013) conducted a snowball search based on the original Manassiev paper, as well as a systematic search with the related topic of "electronic fetal monitoring". We did not find any published work that would interfere with the findings in this manuscript.

Our aim was to first define what one should mean by "normal" fetal heart rate and then to give a data-driven answer to this question, as a basis for the more complicated question about the right choice of "alarm limits".

## MATERIAL AND METHODS

In order to reduce the probability of publishing false positive results, this study followed a strict analysis plan, published before onset of the analyses (*Daumer et al., 2007*). A similar methodology is now being recommended by ENCePP (www.encepp.org) of the European Medical Agency.

### CTG database for exploration and validation

From 2000 to 2007 CTG raw data were systematically collected from three hospitals: the two university hospitals "Technische Universität München" and "Witten-Herdecke" and the non-university hospital of Achern (Germany). "Technische Universität München" and "Witten-Herdecke" are tertiary care centers, while "Achern" is a primary care center. The work program and the corresponding contract were approved by the Department of Obstetrics and Gynecology of the Technische Universität München and the legal department of the Technische Universität München and by the "Ludwig Maximilians University" (cooperation contract in the context of Sonderforschungsbreich SFB 386, subproject B2 Statistische Analyse diskreter Strukturen - Dynamische Modelle zur Ereignisanalyse, from April 28, 2005).

The training data set consisted of the cardiotocograms recorded at "Technische Universität München" from 2000 to 2004. For validation three data sets were used: "Technische Universität München" from 2005 to 2006 for temporal validation, "Witten-Herdecke" from June 2005 to December 2007 and "Achern" from September 2001 to December 2005 for external validation.

We included all 87,510 FHR tracings recorded during the described period on CTG devices linked to the central server in the study, if they were derived from a singleton pregnancy. The included cardiotocograms were obtained both during labor in the delivery room and before onset of labor in the prenatal care unit, starting typically at gestational week 24. The recordings were not necessarily longer than 30 min, as it was originally planned, but a sensitivity analysis (data not shown) suggested, that this did not affect the results. 78,852 tracings demonstrated a sufficient signal quality, necessary for our analysis. For 13,015 CTG tracings collected between 20 and 42 weeks, data about gestational age were available, so that they could be used for analysis of association of FHR and gestational age.

### Investigated variables

For each CTG tracing, the baseline heart rate was extracted from the FHR data coming from the CTG device at a rate of four measurements per second by excluding outlier measurements, eliminating accelerations or decelerations, and averaging based on the "delayed moving windows" algorithm (*Daumer & Neiss, 2001*). These steps were automatically performed by the "Trium CTG Online®" software.

The basis for our analysis was the non-averaged baseline as computed by the CTG online algorithm (*Schindler, 2002*) with one data point as statistical unit.

## Formulation of the normal fetal heart rate range

We considered multiples of five as candidate FHR limits. For this purpose, we first divided the results for the FHR limits by five, rounded to the nearest integer and finally multiplied by five, eventually leading to an approximation of the exact FHR value by an integer ending with 0 or 5 (*Macones et al., 2008*; *National Institute of Child Health and Human Development Research Planning Workshop, 1997*).

We chose the admissible widths of a candidate interval of normal FHR as 40 and 45 bpm. The candidate interval of normal FHR was selected by definition of intervals of 40 or 45 bpm width leading to similar numbers of measurements beyond the lower and upper limit. Further explanations concerning the mathematical optimization problem are provided in the previously published analysis plan (*Daumer et al., 2007*).

## Validation scheme and statistical methodology

By analyzing the "training dataset" a hypothesis for the range of the normal fetal heart rate was built, fulfilling the analysis plan mentioned above. Validation data sets were not opened before the hypotheses were formed. Three independent statisticians did programming of these steps.

# RESULTS

## Patient characteristics

We analyzed 45,915 (Training: 32,325, Validation: 13,590) CTG tracings from the university hospital "Technische Universität München" (2000–2006), 25,294 from the university hospital "Witten-Herdecke" and 7,643 from the non-university hospital of Achern. The pregnant women whose CTG tracings were included were treated antepartum in an in-patient or out-patient setting or they were admitted for delivery (with continuing or intermittent CTG surveillance). Characteristics of the patients delivered during the study period are summarized in Table 1 to give an impression of the population in the respective hospital. They show essentially similar results, but as expected they reveal slight differences consistent with regional characteristics (the small town Achern versus the city of Munich) and the high or low risk collective in tertiary and primary care centers. As an example, older and nulliparous women are more likely to deliver in the university hospitals. Also children with congenital malformations are born preferentially in the University Hospitals, Munich even with a focus on heart malformations as the hospital cooperates with the German Heart Center in Munich for postnatal care of the babies.

A high percentage of the tracings were obtained ante partum or from women during first stage of labor as, for example, in "Technische Universität München" only 7,465 women (16.2% of tracings) were delivered under CTG surveillance in the years of 2000 to 2006, while 45,915 CTG tracings were recorded. In "Witten-Herdecke" 3,527 women (13.9%) were delivered and 25,294 CTG tracings were recorded, in "Achern" there were 1,788 deliveries (23.4%), but 7,643 CTG tracings were recorded. Our study comprises all weeks of pregnancies with analyzable CTG tracings, typically starting at 24 completed gestational

**Table 1 Patient characteristics.** Description of patient characteristics.

| | Characteristics | | Training TUM 2000–2004 n (%) | Validation I TUM 2005–2006 n (%) | Validation II WH 06/2005–2007 n (%) | Validation III A 09/2001–2005 n (%) |
|---|---|---|---|---|---|---|
| | Number of delivered women | | 5,366 | 2,323 | 3,542 | 1,788 |
| | Cardiotocogram recorded during delivery | | 5,184 (96.6) | 2,281 (98.2) | 3,527 (99.6) | n/a |
| | Maternal age | <20 J. | 88 (1.6) | 38 (1.6) | 105 (3.0) | 78 (4.5) |
| | | 20–29 J. | 1,707 (31.9) | 744 (32.0) | 1,440 (40.7) | 739 (42.6) |
| Mother | | 30–39 J. | 3,249 (60.8) | 1,371 (59.0) | 1,857 (52.4) | 866 (49.9) |
| | | ≥ 40 J. | 302 (5.6) | 169 (7.3) | 140 (4.0) | 51 (2.9) |
| | Nulliparous women | | 2,387 (44.7) | 986 (42.5) | 1,477 (41.7) | 458 (27.9) |
| | Gestational age at delivery | MW ± STD | 38.3 ± 3.0 | 38.2 ± 3.0 | 38.4 ± 2.4 | 38.8 ± 3.0 |
| | Normal delivery | | 3,058 (57.1) | 1,237 (53.3) | 1,992 (56.2) | 1,050 (58.4) |
| | Forceps extraction | | 88 (1.6) | 14 (0.6) | 75 (2.1) | 0 (0) |
| Delivery | Vacuum extraction | | 263 (4.9) | 131 (5.6) | 71 (2.0) | 137 (7.6) |
| | Elective Cesarean | | 824 (15.4) | 405 (17.4) | 774 (21.9) | 289 (16.1) |
| | Secondary Cesarean | | 1,118 (20.9) | 535 (23.0) | 630 (17.8) | 321 (17.9) |
| | Tocolysis during delivery | | 1,177 (21.9) | 584 (25.2) | 645 (18.2) | n/a |
| | Male | | 2,799 (52.2) | 1,177 (50.7) | 1,799 (50.2) | 927 (51.8) |
| | Female | | 2,567 (47.8) | 1,146 (49.3) | 1,743 (49.8) | 861 (49.2) |
| Fetal outcome | Birthweight (g) | MW ± STD | 3,157 ± 727 | 3,138 ± 731 | 3,263 ± 631 | 3,393 ± 475 |
| | Congenital malformation[a] | | n/a | 75 (3.2) | 125 (3.5) | 15 (0.8) |
| | Congenital heart malformation[a] | | n/a | 36 (1.5) | 11 (0.3) | n/a |

**Notes.**

n/a, Data not available or quality not sufficient.

[a] Via ICD-10 coding.

weeks. But more than 75 percent of the CTG tracings were obtained from pregnancies older than 37 weeks.

## Fetal heart rate analysis

The distribution of the FHR baseline measurements of the training data set over the whole range of possible frequencies is shown as a histogram in Fig. 1A, showing roughly the shape of a Gaussian distribution, but not the full symmetry. Distribution in steps of 5 bpm is summarized in Table 2 as a percentage of all measurements for the training data (Column 1).

The criterion for definition of the best interval is

$$\arg \min_{i=1,\dots,5} (\widehat{F}(Z_{lower}^{(i)}) - (1 - \widehat{F}(Z_{upper}^{(i)})))^2.$$

(for further details see our analysis plan (*Daumer et al., 2007*)).

Analyzing the training set, the selected interval of 40 to 45 bpm width was 115 to 160 bpm (criterion: $(0.0181 - 0.0321)^2 = 0.20 \cdot 10^{-3}$). The criterion for the interval with

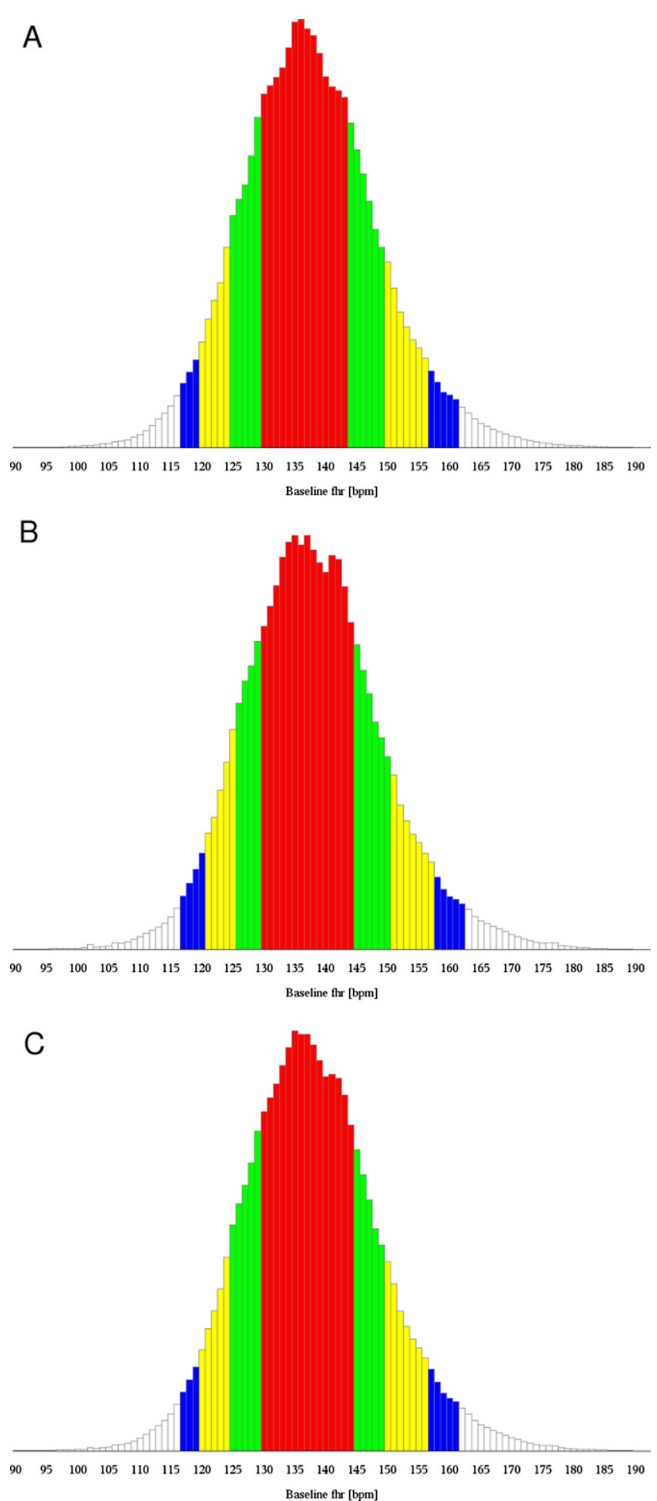

**Figure 1 Histogram of baseline fetal heart rate values** (A) Training data. (B) Validation data. (C) All data. Red bars comprise 25th to 75th percentile, red and green ones 12.5th to 87.5th percentile, red, green and yellow bars 5th to 95th percentile and all bars except white ones comprise 2.5th to 97.5th percentile.

**Table 2  Distribution of the fetal heart rate in the training and validation sets.** The number of singular fetal heart rate recordings under or above the given limits of fetal heart rate as a percentage of all measurements is displayed.

|  | Training TUM 2000–2004 | Validation I TUM 2005–2006 | Validation II WH 06/2005–2007 | Validation III A 09/2001–2005 | Validation I - III |
|---|---|---|---|---|---|
| **Lower limit** | | | | | |
| <100 bpm | 0.13% | 0.15% | 0.08% | 0.17% | 0.12% |
| <105 bpm | 0.26% | 0.26% | 0.15% | 0.37% | 0.24% |
| <110 bpm | 0.62% | 0.64% | 0.40% | 0.78% | 0.57% |
| <115 bpm | 1.81% | 1.79% | 1.24% | 1.68% | 1.53% |
| <120 bpm | 5.02% | 4.90% | 3.54% | 4.45% | 4.21% |
| **Upper limit** | | | | | |
| >145 bpm | 23.26% | 23.81% | 27.84% | 22.33% | 25.22% |
| >150 bpm | 12.56% | 13.13% | 16.09% | 12.04% | 14.16% |
| >155 bpm | 6.51% | 6.96% | 8.67% | 6.23% | 7.53% |
| >160 bpm | 3.21% | 3.55% | 4.35% | 3.11% | 3.79% |
| >165 bpm | 1.47% | 1.76% | 2.00% | 1.51% | 1.80% |
| >170 bpm | 0.68% | 0.78% | 0.92% | 0.70% | 0.82% |

**Table 3  Calculation of the criterion for definition of the best interval in the training and validation data sets.** Square of difference between upper and lower tail of the distribution ([i]), as shown in Table 3. All values have to be multiplied with $10^{-3}$. The best criterion for each data set is marked in bold letters.

|  | Training TUM 2000–2004 | Validation I TUM 2005–2006 | Validation II WH 06/2005–2007 | Validation III A 09/2001–2005 | Validation I - III |
|---|---|---|---|---|---|
| 110–150 | 14.24 | 15.60 | 24.62 | 12.69 | 18.48 |
| 110–155 | 3.46 | 3.99 | 6.83 | 2.97 | 4.85 |
| 115–155 | 2.21 | 2.68 | 5.51 | 2.07 | 3.61 |
| **115–160** | **0.20** | 0.31 | 0.97 | 0.20 | 0.51 |
| **120–160** | 0.33 | **0.18** | **0.07** | **0.18** | **0.02** |
| 120–165 | 1.26 | 0.98 | 0.24 | 0.86 | 0.58 |

120 to 160 bpm was only marginally bigger (criterion: $(0.0502 - 0.0321)^2 = 0.33 \cdot 10^{-3}$) (Table 4, Column 1), such that the lower bound, in contrast to the upper bound, is not stable.

Hence the following hypotheses were formulated and tested during validation:

1. The upper limit of the FHR should be 160 bpm.
2. The lower limit should be either 115 or 120 bpm.

Results of each of the validation data sets and of a combination of all three of them revealed the range of 120 to 160 bpm as the best interval (Fig. 1B, Tables 2 and 3, Columns 2, 3, 4, and 5). Hence, both hypotheses were validated.

**Table 4 Distribution of FHR baseline during gestation.** (A) 95% confidence intervals for mean FHR baseline are displayed for intervals of several gestational weeks. All pairwise comparisons are significant ($p < 0.01$) with both t-test and Mann-Whitney tests. The comparisons between gestational age of $>= 37$ and other groups are the most significant. (B) 95% confidence intervals for mean FHR baseline within the group of gestational age of 37 weeks or more.

| Gestational age | n | 95% confidence interval | | |
|---|---|---|---|---|
| **A** | | | | |
| <28 | 1230 | 140.7538 | – | 141.9422 |
| 28 – <32 | 1059 | 139.1587 | – | 140.3843 |
| 32 – <37 | 2248 | 138.1575 | – | 138.9322 |
| >=37 | 8478 | 136.0104 | – | 136.4295 |
| **B** | | | | |
| 37 | 1090 | 136.7176 | – | 137.8588 |
| 38 | 1793 | 135.5575 | – | 136.4720 |
| 39 | 1962 | 135.9786 | – | 136.8404 |
| 40 | 2325 | 135.2181 | – | 136.0158 |
| 41 | 1199 | 135.9135 | – | 137.0438 |
| 42 | 109 | 133.2492 | – | 137.8009 |

The mean FHR baseline plotted against gestational age is shown in Fig. 2. Table 4 shows 95% confidence intervals for mean FHR baseline in different gestational weeks. Regression analysis with the median FHR baseline as dependent variable and the gestational age (in weeks) as independent variable yielded a slope estimate of $-0.378$ ($p < 0.001$), meaning that the median FHR decreases on average by 0.4 bpm per week of pregnancy. The assumptions underlying the linear regression model were approximately fulfilled.

## DISCUSSION

Analyzing about 1.5 billion individual single baseline fetal heart rate measurements from 78,852 CTG tracings in three German medical centers, we found that "normal" ranges – normality in a statistical sense - are 120 to160 bpm. By this data-driven definition of the normal FHR we aimed to generate a solid basis for the clinically important attempt to eventually further reduce the rate of false alarms in CTG monitoring in general and electronic decision support systems in particular. This might help to avoid unnecessary interventions such as Cesarean sections. The FHR baseline in our analysis decreases slightly during gestation, in line with results of other groups (*Nijhuis et al., 1998*; *Serra et al., 2009*). There are well-known physiological changes in fetal development that are consistent with this empirical finding (*Karolina & Edwin, 2011*), essentially due to the increasing opposed effect of the sympathetic nervous system as gestational age increases.

Validation of the results in an independent data set is a crucial step to avoid the publication of false positive research findings (*Daumer et al., 2008*; *Ioannidis, 2005*). Both temporal validation (based on data collected later than the training data) and external validation (based on data collected in another medical center), used in our study, are known to be essential (*König et al., 2007*). Furthermore, the strict blind validation

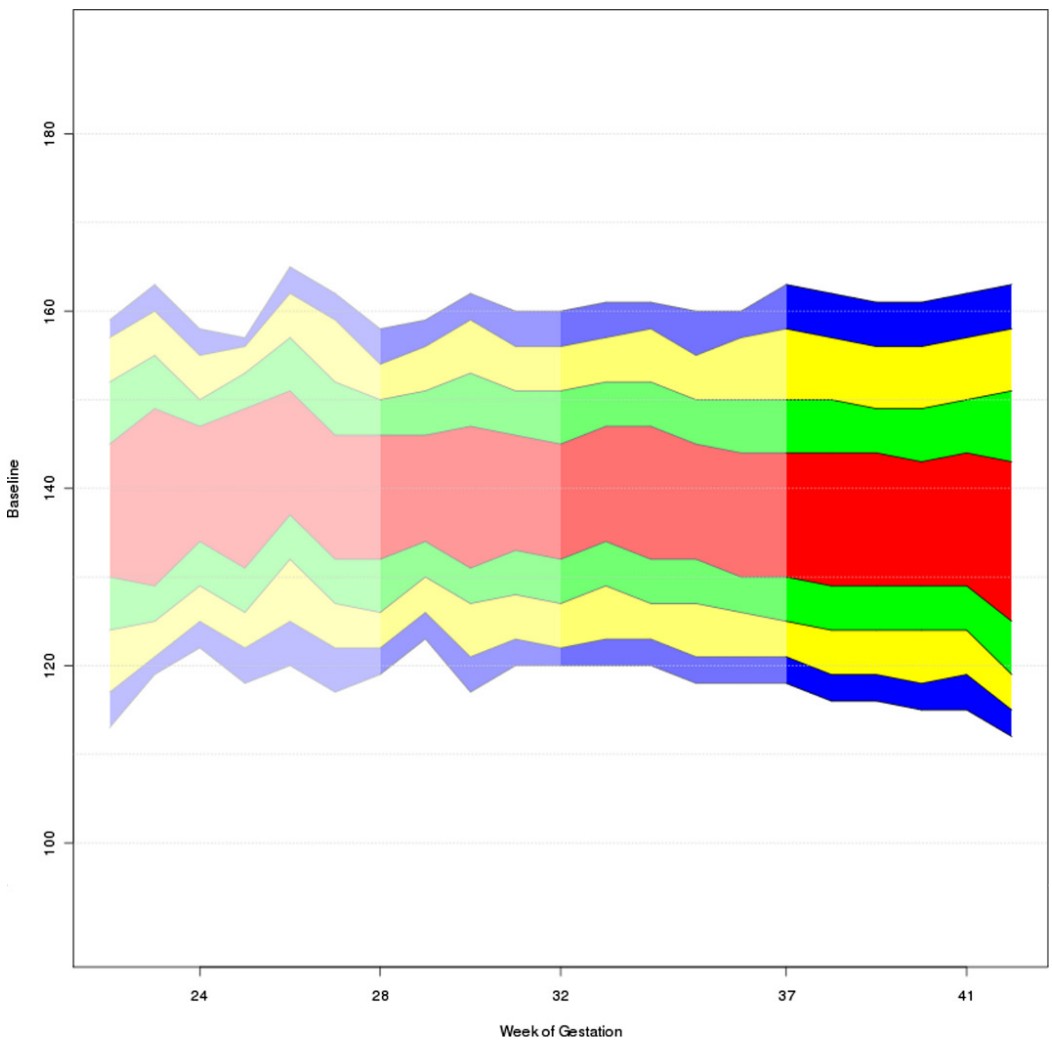

**Figure 2 Quantile bands of FHR plotted against gestational age.** FHR (bpm) is plotted against gestational weeks from 20 to 42. Red colours comprise 25th to 75th percentile, red and green colours 12.5th to 87.5th percentile, red, green and yellow colours 5th to 95th percentile and all colours comprise 2.5th to 95.5th percentile.

procedure was adopted and described in a detailed analysis plan in the pre-publication platform *Nature Precedings* (*Daumer et al., 2007*) before starting the analyses. The results about the normal range are very robust, indicating that neither the type of hospital which is potentially linked to special selection criteria for the pregnant women nor the time as measured roughly in 5–10 year intervals seems to play a role – an argument for the external validity of the findings in the exploratory part.

For user acceptance we used steps of 5 bpm as possible borders of the normal FHR as recommended in the consensus meeting of the National Institute of Child Health and Human Development (*Macones et al., 2008*; *National Institute of Child Health and Human Development Research Planning Workshop, 1997*). The width of the interval of 40 to 45 bpm

was traditionally used in many international guidelines. As we planned the study, we chose no other intervals, as narrowing of the interval would increase the false alarm rate and wider intervals could miss pathologic conditions of the fetus.

The upper limit of 160 bpm raised concerns in the FIGO meeting in 1985, as Saling described abnormal findings in 24% of scalp blood analyses if the baseline was higher than 160 bpm (*Saling, 1966*). It could be shown that the current FIGO guidelines based on computerized analyses of the CTG show a high sensitivity to detect fetal acidosis in case of a suspect or pathological classification of the baseline level. It may turn out that a modification of the normal ranges further improves sensitivity and specificity of fetal acidosis during labor (*Schiermeier et al., 2008*). Also, multivariate modeling involving fetal and maternal outcome data may improve evidence-based online decision support tools.

Data from a recently published study in a different context (*Serra et al., 2009*) is compatible with the findings of our exploratory analysis with a lower limit of 115 or 120 bpm for the gestational ages. Data for the 97th and 99th percentiles are not shown in this study. But shifting the lower limit to 120 will increase the number of false alarms whereas a lower limit of 115 will inevitably increase the risk to misinterpret maternal heart rates as fetal heart rate. This last problem has raised many concerns and discussions about technical solutions for differentiation of maternal and fetal heart rate, as fatal consequences for the fetus could occur (*Murray, 2004*). The new German guideline (*Deutsche Gesellschaft für Gynäkologie und Geburtshilfe, 2012*) recommends therefore simultaneous recording of fetal and maternal heart rate, technically possible either by maternal pulse oxymetry integrated in a CTG device or simultaneous ECG recording of mother and fetus.

As FHR tracings of prenatal care patients were included, our study population consists of a fraction of pregnancies remote from term, eventually resulting in higher baselines as suggested before. As our analysis according to gestational ages shows, the upper limit of 160 bpm is valid for younger and for later gestational ages. A lower limit of 120 bpm leads only near term to more false alarms since normal FHR decreases further, and is more appropriate, as discussed above, to avoid misinterpretation of maternal heart beat as FHR. There are no different guidelines for scoring cardiotocograms of early gestational ages as this would be too difficult in daily practice. Only computerized algorithms could use boundaries without rounding based on multivariate modeling and correlate these results to fetal outcome.

FIGO guidelines defined boundaries from 110 to 150 bpm, representing the approximately 0.6th to 86th percentile from our study. Current guidelines released by the American College of Obstetricians and Gynecologists (*American Congress of Obstetricians and Gynecologists, 2009*), the National Institute of Child Health and Human Development (*National Institute of Child Health and Human Development Research Planning Workshop, 1997*), the Society of Obstetricians and Gynaecologists of Canada (*Society of Obstetrics and Gynaecologists of Canada, 2007*), the United Kingdom's National Institute for Health and Clinical Excellence (*National Institute for Health and Clinical Excellence (NICE), 2007*), the Royal Australian and New Zealand College of Obstetricians and Gynaecologists (*Royal Australian and New Zealand College of Obstetricians and Gynaecologists, 2006*) and the

Japan Society of Obstetrics and Gynecology (*Perinatal Committee of the Japan Society of Obstetrics and Gynecology, 2009*) define a very wide range of normal FHR with 110 to 160 bpm, representing the approximately 0.6th to 96th percentile. We raised concerns about the broad width of the range of 50 bpm and the lower limit of 110 bpm. As these guidelines are in use for some years in many countries at the moment, we assume that this range is still safe for detection of fetal compromise. In contrast, specificity of the CTG for fetal acidosis becomes better. But safety-analyses should confirm this assumption.

Our results have stimulated discussions within the corresponding German society "Deutsche Gesellschaft für Gynäkologie und Geburtshilfe" (*Deutsche Gesellschaft für Gynäkologie und Geburtshilfe, 2010*) having led to a recent update of the previous guidelines (*Deutsche Gesellschaft für Gynäkologie und Geburtshilfe, 2012*), based on data from the exploratory analysis. We hope that our study will trigger a process of continuous improvement of evidence based clinical decision making in fetal monitoring – perhaps a task to be triggered by the HTA working group of ENCePP (http://www.encepp.eu/structure/documents/ENCePPWGHTA_Mandate.pdf).

## ACKNOWLEDGEMENTS

We thank Nicholas Lack from the "Bayerische Arbeitsgemeinschaft für Qualitätssicherung" and Thomas Füsslin, Ortenau Klinikum Achern, for their support in providing information about the pregnancies at the Klinikum rechts der Isar and Ortenau Klinikum Achern. We thank Nadja Harner, Martina Günter and Michael Scholz for data management and technical support. We also would like to thank Erich Saling for helpful discussions and the speaker of Biomed-S and former speaker of the DFG-funded Sonderforschungsbereich SFB386 Prof. Dr. Fahrmeir, Ludwig-Maximilians University, for continuous support. The comments of Marlene Sinclair and another anonymous reviewer have helped to further improve the manuscript. The authors thank the Porticus Foundation for their generous support for the International School for Clinical Bioinformatics & Technical Medicine.

### Funding

There was no funding for the study or for publication, but the Sylvia Lawry Centre for Multiple Sclerosis Research, Munich, Germany, has received support from the Porticus Foundation in the context of the "International School for Clinical Bioinformatics and Technical Medicine".

### Grant Disclosures

The following grants were received by Martin Daumer as professional support during the time of the study but were not directly for use in this study:
NETSIM - European Union FP7: Grant No 215820.
VPHOP - European Union FP7: Grant No 223865.
ABMA - Federal Ministry of Economics and Technology: Grant No KF0564001KF7.
EBDiMS - Hertie Foundation: Grant No 1.01.1/07/015.

Intl. School for Clinical Bioinformatics - Porticus Foundation: Grant No 900.50578.
Cardiogenics - European Union FP6: Grant No LSHM-CT-2006-037593.
Bloodomics - European Union FP6: Grant No LSHM-CT-2004-503485.

The following grant was received by Dr. Christian Lederer & Dr. Anne-Laure Boulesteix as partial support when employed at the SLCMSR as fellows of the International School for Clinical Bioinformatics & Technical Medicine: Grant No 90050578.

## Competing Interests

Martin Daumer is Director of the Sylvia Lawry Centre for MS Research. He is also one of the two managing directors of Trium Analysis Online GmbH, together with Michael Scholz (50% ownership each). Trium is a manufacturer of CTG monitoring systems.
Dr. Daumer serves on the scientific advisory board for the EPOSA study; has received funding for travel from ECTRIMS; serves on the editorial board of MedNous; is co-author with Michael Scholz on patents re: Apparatus for measuring activity (Trium Analysis Online GmbH), method and device for detecting a movement pattern (Trium Analysis Online GmbH), device and method to measure the activity of a person (Trium Analysis Online GmbH), co-Author with Christian Lederer of device and method to determine the fetal heart rate from ultrasound signals (Trium Analysis Online GmbH), author of method and device for detecting drifts, jumps and/or outliers of measurement values, coauthor of patent applications with Michael Scholz of device and method to determine the global alarm state of a patient monitoring system, method of communication of units in a patient monitoring system, and system and method for patient monitoring; serves as a consultant for University of Oxford, Imperial College London, University of Southampton, Charite, Berlin, University of Vienna, Greencoat Ltd, Biopartners, Biogen Idec, Bayer Schering Pharma, Roche, and Novartis; and receives/has received research support from the EU-FP7, BMBF, BWiMi, and Hertie Foundation.
Nadja Harner was an employee of Trium, Anne-Laure Boulesteix was an employee of the SLC when the study was conducted.
There is no known financial or other conflict of interests for the other authors.

## Author Contributions

- Stephanie Pildner von Steinburg conceived and designed the experiments, performed the experiments, analyzed the data, wrote the paper.
- Anne-Laure Boulesteix and Martin Daumer conceived and designed the experiments, analyzed the data, contributed reagents/materials/analysis tools, wrote the paper.
- Christian Lederer analyzed the data, contributed reagents/materials/analysis tools, critical review of manucript.
- Stefani Grunow analyzed the data, contributed reagents/materials/analysis tools.
- Sven Schiermeier performed the experiments, analyzed the data, wrote the paper.
- Wolfgang Hatzmann performed the experiments, and critical review of mansucript.

- Karl-Theodor M. Schneider conceived and designed the experiments, performed the experiments, and critical review of manuscript.

## Human Ethics

The following information was supplied relating to ethical approvals (i.e. approving body and any reference numbers):

The work program and the corresponding contracts were approved by the Department of Obstetrics and Gynecology of the Technische Universität München and the legal department of the Technische Universität München, and by the Ludwig Maximilians University (cooperation contract in the context of Sonderforschungsbreich SFB 386, subproject B2 "Statistische Analyse diskreter Strukturen - Dynamische Modelle zur Ereignisanalyse, from April 28, 2005).

## Patent Disclosures

The following patent dependencies were disclosed by the authors:

Martin Daumer is the inventor of: method and device for detecting drifts, jumps and/or outliers of measurement values, US Patent 6,556,957, April 29, 2003, German Patent application Nr. 198 39 047.5-35, Nov. 11, 2005, European Patent 1097439 (99939929.8-2215), March 3, 2004.

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
