# Peer review of "What is the “normal” fetal heart rate?"

_PeerJ, doi:10.7717/peerj.82_

## Round 0.1 · original submission · Minor Revisions

We are willing to proceed with further review if you address the following:
1. Reanalyze and describe the time dependent norms for gestational HR as suggested by the reviewer
2. Please develop a clear explanation for the methods to reassure the readers that all boundries of the available literature have been reviewed
3. Reduce the writing in the sections that deviate from the main focus of the study or with techniques

·

Basic reporting

see below

Experimental design

see below

Validity of the findings

see below

Additional comments

The paper conforms to your standard submission guidelines and the only issue I have is the missing raw data. However, the author has provided an explanation and this is reasonable.
The paper provides new knowledge about an important fetal heart rate parameter for normality and challenges the status quo based on data collection from 3 German medical centers from 2000-2007 (78,852 tracings of sufficient quality). The value of the paper for discussion on raising CTG alarms when the FH is outside these ‘normal parameters’ is key.

I have two issues. The literature search strategy and process is not clear and this could lead to an omission of key papers not included in the COCHRANE review and I think the authors should consider how best to deal with convincing the reader that their literature review is indeed of the highest quality.

Changing current practice and changing CTG specifications will require a model or framework for adoption of change and this would enhance the citation of the paper.

The paper is worthy of publication with attention to the above minor modifications and I do not need to see the paper again.

·

Basic reporting

A retrospective study on a large pregnant subject population to determine the normal range of fetal heart rate from 20 to 42 weeks of gestation

Experimental design

Appropriate study design with the use of a valid and proven method that was previously published in peer reviewed medical journals.

Validity of the findings

Appropriate analysis and interpretation of the data.

Additional comments

Dr. Stephanie Pildner von Steinburg and associates conducted a retrospective study to determine the range of normal fetal heart rate during gestation, from 20 to 42 weeks. The study involved large population of subjects with an established, and previously published, method of data analysis. They concluded that the range of normal fetal heart rate during gestation is 120 to 160 BPM. Not surprisingly, the range of fetal heart rate described here is the same as was described originally by Dr. Edward Hon, one of the pioneer in electronic fetal rate monitoring who also developed classifications for normal as well as abnormal fetal heart rate pattern.

Although this study re-confirms generally accepted previous norms, it disputes the validity of other observations that suggest different baseline fetal heart rate.

Other comments: The main subject of this study is to establish a normal baseline fetal heart rate but in the section of introduction the authors bring up the subject of fetal cardiotocography, its clinical usefulness in reducing perinatal morbidity and mortality. The authors should limit their manuscript related to baseline fetal heart rate rather than the usefulness or potential benefits of fetal cardiotocography. Therefore, the section of introduction should start with the page 3, line 60 “currently, there is not even agreement on the normal range on the baseline of the FHR.

The authors state that the baseline fetal heart rate decreases during the course of gestation and in figure 2 they show a graph related to it. A short explanation why the fetal baseline heart rate during advancing gestation decreases should be given. Explanation: During the early gestation sympathetic nervous system dominates over parasympathetic nervous system. With advancing gestation age towards term gestation vagal tone become more dominant with a decrease in baseline fetal heart rate. Since the authors have a large number of subjects in their study, they could stratify subject population in different gestational groups, i.e. 20 to 24, 25 to 28, 20 to 32 weeks, etc. in order to establish norms for baseline fetal heart rate for different gestation age.

---

## Round 0.2 · accepted · Accept

I appreciate your attention to our request to outline the review boundaries as well as the new changes to define the temporal change in heart rates. When you review the proofs, please be sure to expand Dr. Lederer's first name in the title.